# Early events in amyloid-β self-assembly probed by time-resolved solid state NMR and light scattering

Jaekyun Jeon[1,2], Wai-Ming Yau[1] & Robert Tycko [1] ✉

Self-assembly of amyloid-β peptides leads to oligomers, protofibrils, and fibrils that are likely instigators of neurodegeneration in Alzheimer's disease. We report results of time-resolved solid state nuclear magnetic resonance (ssNMR) and light scattering experiments on 40-residue amyloid-β (Aβ40) that provide structural information for oligomers that form on time scales from 0.7 ms to 1.0 h after initiation of self-assembly by a rapid pH drop. Low-temperature ssNMR spectra of freeze-trapped intermediates indicate that β-strand conformations within and contacts between the two main hydrophobic segments of Aβ40 develop within 1 ms, while light scattering data imply a primarily monomeric state up to 5 ms. Intermolecular contacts involving residues 18 and 33 develop within 0.5 s, at which time Aβ40 is approximately octameric. These contacts argue against β-sheet organizations resembling those found previously in protofibrils and fibrils. Only minor changes in the Aβ40 conformational distribution are detected as larger assemblies develop.

Self-assembly of amyloid-β (Aβ) peptides has been studied intensively since the initial identification of Aβ as the primary proteinaceous component of amyloid plaques in Alzheimer's disease (AD) brain tissue more than 35 years ago[1,2]. Starting from a primarily monomeric state at 15–40 μM, photochemical crosslinking experiments indicate that Aβ oligomers containing at least seven molecules can form within about 100 s[3]. Larger metastable assemblies, including globular oligomers[4–7] and worm-like protofibrils[8–10], can then form on the time scale of minutes to hours before mature fibrils with lengths exceeding 1 μm are observed. At low concentrations, formation of globular oligomers and protofibrils can be avoided[11], leading to a more direct progression from monomers to fibrils with kinetics on the time scale of hours dictated by primary nucleation followed by fibril fragmentation or surface-catalyzed secondary nucleation, depending on experimental conditions[12,13].

The monomeric state of full-length 40- and 42-residue Aβ peptides (Aβ40 and Aβ42) is primarily unstructured under conditions that disfavor aggregation[14–16]. Although the fibrillar state is polymorphic, with multiple distinct structures for Aβ40 and Aβ42 fibrils having been characterized by solid state nuclear magnetic resonance (ssNMR)[17–23] and cryogenic electron microscopy (cryo-EM) methods[24–27], the core structure within a given fibril is homogeneous. Within all mature fibril polymorphs, Aβ peptides adopt conformations that include multiple β-strand segments, which form in-register parallel β-sheets through intermolecular hydrogen bonds among backbone groups[28–31]. According to ssNMR data, Aβ conformations in large oligomers[5,7,32–34] and protofibrils[10,34–36] resemble those in mature fibrils, albeit with reduced overall conformational order. However, β-sheet structures in large oligomers and protofibrils can be qualitatively different from those in mature fibrils[6,7,10,34,36,37].

In the initial stages of Aβ self-assembly, oligomers comprising an increasing number of molecules develop as short-lived intermediates (Fig. 1a) before the large globular oligomers, protofibrils, and fibrils that are readily observable in transmission electron microscope (TEM) images (Fig. 1b–d and Supplementary Fig. 1). Molecular structural properties of early intermediates in Aβ self-assembly have not been established by experiments to date. For example, although it is known that large oligomers (*n* » 10, where

[1]Laboratory of Chemical Physics, National Institute of Diabetes and Digestive and Kidney Diseases, National Institutes of Health, Bethesda, MD 20892-0520, USA. [2]Institute for Bioscience and Biotechnology Research, University of Maryland/National Institute of Standards and Technology, Rockville, MD 20850, USA. ✉e-mail: robertty@mail.nih.gov

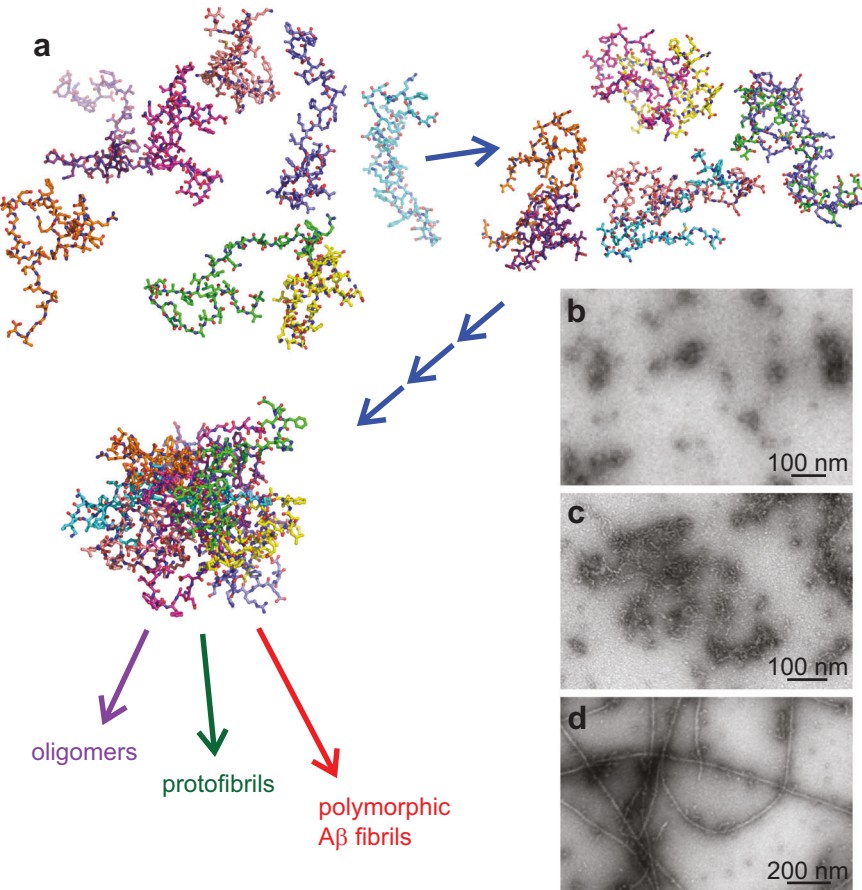

**Fig. 1 | Hypothetical depiction of Aβ40 self-assembly.** After a rapid change from solvent conditions that favor the monomeric state to conditions that favor self-assembly, Aβ40 monomers that are initially conformationally disordered form oligomers with progressively larger sizes (**a**), possibly with increasingly ordered molecular and supramolecular structures. Within minutes, metastable nonfibrillar oligomers (**b**) and protofibrillar assemblies (**c**) are readily apparent in negative-stain TEM images. After longer incubation periods, thermodynamically stable amyloid fibrils (**d**) become the predominant self-assembled state. Images in **b**–**d** are representative examples from sets of 19, 32, and 18 micrographs, respectively.

$n$ is the number of Aβ molecules), protofibrils, and fibrils have β-strand-rich conformations, the stage of self-assembly at which β-strand conformations develop is not known experimentally. How the degree of structural order, including homogeneity of molecular conformations and intermolecular alignments, depends on oligomer size is also not known from experiments. The transient nature and heterogeneity of early intermediates in Aβ self-assembly renders standard methods for molecular structure determination largely inapplicable.

To address problems of this type, we have recently developed time-resolved ssNMR methods[38,39], based on a combination of rapid mixing to initiate a structural evolution process, rapid freezing to trap intermediate states, and low-temperature ssNMR technology with sensitivity enhancements from dynamic nuclear polarization (DNP)[40,41] to enable efficient measurements on frozen peptide and protein solutions with concentrations around 1 mM. Here we apply time-resolved ssNMR to Aβ40 self-assembly, initiated by a rapid pH drop. We combine the ssNMR data with data from time-resolved light scattering measurements, which allow us to quantify the evolution of oligomer size distributions on time scales from 5 ms to many minutes. Based on the combined data, we find that β-strand-rich conformations develop very rapidly, within 1 ms of the pH drop, at which time Aβ40 molecules are still primarily monomeric on average. Modeling of the light scattering data indicates oligomer growth up to sizes of approximately 8 molecules within 0.5 s under our experimental conditions, followed by a more gradual and nearly linear increase in average size that can be modeled as a coagulation process. Although the average value of $n$ is approximately 50 after 10 min in our experiments, the time-resolved ssNMR data show surprisingly little change in conformational preferences and overall structural order at the molecular level as the nonfibrillar Aβ40 assemblies grow. These data also suggest early formation of intramolecular contacts that are independent of $n$, consistent with U-shaped[10,17,19,30] or hairpin-like[26,36,42] molecular conformations, and a more gradual development of intermolecular contacts that differ from those in Aβ40 fibrils.

## Results

### Initiation of Aβ40 self-assembly by a rapid pH drop

As depicted in Fig. 2a, time-resolved ssNMR experiments began with isotopically labeled, synthetic Aβ40 solutions in 20 mM NaOH (pH ≈ 12), where Aβ40 is fully soluble and monomeric at 2.3 mM. Self-assembly was initiated by mixing Aβ40 solutions in a 2:1 ratio with 525 mM sodium phosphate buffer in 0.7–3.0 ms (depending on flow rate and mixer volume, see Methods), thereby dropping the pH value to 7.4 and producing a final Aβ40 concentration of 1.5 mM. After structural evolution times $\tau_e$ from 0.7 ms to 1.0 h, solutions were frozen in less than 0.5 ms[38] by spraying a high-speed jet (0.85–2.6 cm/ms from a 50 μm diameter nozzle at 1.0–3.0 ml/min flow rates) onto a rotating copper plate that was pre-cooled to 77 K in liquid nitrogen. Frozen material was then packed into magic-angle spinning (MAS) ssNMR rotors under liquid nitrogen and stored at 77 K. The home-built apparatus for rapid mixing and freeze-trapping has been described previously[38]. Values of $\tau_e$ from 0.7 ms to 1.0 h were achieved by varying

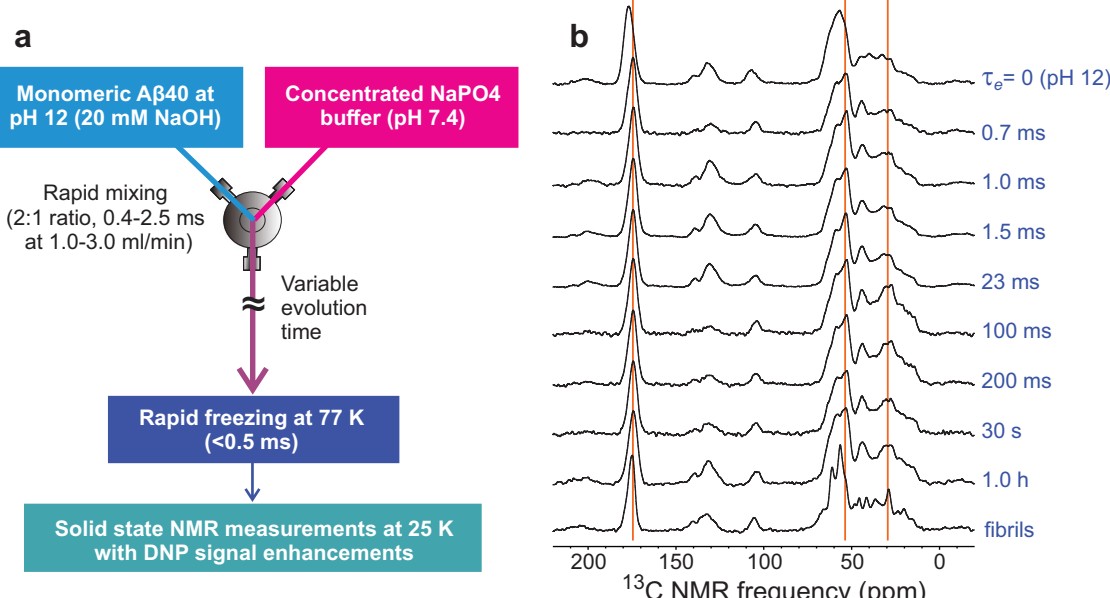

**Fig. 2 | Strategy for time-resolved ssNMR studies of Aβ40 self-assembly. a** An Aβ40 solution at pH 12 is rapidly mixed with a concentrated pH 7.4 buffer to initiate the process. The mixed solution is rapidly frozen on a cold copper surface after a structural evolution period $\tau_e$, which is controlled by the distance and/or volume between the mixer and the cold plate and by the flow rate. Structural information is obtained from low-temperature, DNP-enhanced ssNMR measurements on the frozen solutions. **b** Double-quantum-filtered 1D $^{13}$C ssNMR spectra of frozen solutions with [Aβ40] = 1.5 mM and the indicated values of $\tau_e$. Aβ40 was $^{13}$C-labeled at all carbon sites of F19, V24, G25, S26, A30, I31, L34, and M35 (Aβ40-FVGSAILM). Fibrils with the same isotopic labeling pattern were prepared separately before rapid freezing. Vertical orange lines indicate some of the positions where $\tau_e$-dependent changes in the spectra are evident.

the flow rates, the mixer volume, the volume between the mixer and the jet nozzle, and the distance from the nozzle to the cold copper surface (see "Methods" and Supplementary Table 1).

Figure 2b shows one-dimensional (1D) $^{13}$C ssNMR spectra of frozen Aβ40 solutions with various values of $\tau_e$. Spectra were recorded with DNP at sample temperatures of 25 K[41], using 10 mM sulfoacetyl-DOTOPA[43] as the paramagnetic dopant; double-quantum filtering[44] was used to suppress residual signals from glycerol, which was included as a cryoprotectant (see "Methods"). Circular dichroism spectra indicate that addition of glycerol does not alter the conformational properties of Aβ40 substantially (Supplementary Fig. 2). For these spectra, Aβ40 was $^{13}$C-labeled at all carbon sites of eight residues, namely F19, V24, G25, S26, A30, I31, L34, and M35 (Aβ40-FVGSAILM). Large changes in peak positions and lineshapes are observed between $\tau_e = 0$ (rapidly frozen at pH 12 without a pH drop) and $\tau_e = 0.7$ ms. From $\tau_e = 0.7$ ms to $\tau_e = 1.0$ h, spectral changes are subtle, consisting of a growth of intensity in the 25–35 ppm region up to 100 ms. The 1D $^{13}$C ssNMR spectrum of A40-FVGSAILM fibrils, prepared by seeded growth and frozen after the addition of glycerol and DNP dopant (see Methods), is qualitatively different, with sharper features that indicate a higher level of structural order.

**Evolution of secondary structure from time-resolved 2D solid state NMR**

Figure 3a shows examples of two-dimensional (2D) $^{13}$C ssNMR spectra of the frozen A40-FVGSAILM solutions with various values of $\tau_e$. These 2D spectra were obtained with $^{13}$C–$^{13}$C spin diffusion mixing periods $\tau_{sd}$ equal to 20 ms, producing strong intra-residue (but not inter-residue) crosspeaks. Although crosspeaks are broad and overlapping, clear changes in positions of intensity maxima are observed between $\tau_e = 0$ and $\tau_e = 0.7$ ms, some of which are indicated by the cyan and gold lines in Fig. 3a. From $\tau_e = 0.7$ ms to $\tau_e = 1.0$ h, no clear changes in intensity patterns are observed. The 2D spectrum of fibrillar Aβ40-FVGSAILM is qualitatively different, with sharper crosspeaks and somewhat different crosspeak positions. The full set

of 2D spectra and representative 1D slices are shown in Supplementary Figs. 3 and 4.

To quantify changes in crosspeak intensity patterns, pairwise root-mean-squared deviation (rmsd) values were calculated after normalizing the intensities in each 2D spectrum to the total crosspeak volumes within the relevant spectral regions. Results are displayed as heat maps in Fig. 3b, c for aliphatic-aliphatic and aliphatic-carbonyl regions, respectively. These analyses confirm that differences among 2D spectra of A40-FVGSAILM samples with 0.7 ms ≤ $\tau_e$ ≤ 1.0 h are not significantly above the noise levels in these spectra (rmsd values of 0.27 ± 0.13 and 0.24 ± 0.12 in Fig. 3b, c, respectively; reported as average ± standard deviation). 2D spectra of the sample with $\tau_e = 0$ and the fibrillar sample are significantly different from spectra of samples with 0.7 ms ≤ $\tau_e$ ≤ 1.0 h (rmsd values of 0.75 ± 0.26 and 0.87 ± 0.09 for the sample with $\tau_e = 0$ in Fig. 3b, c, respectively; rmsd values of 0.94 ± 0.19 and 0.51 ± 0.06 for the fibrillar sample in Fig. 3b, c, respectively).

Figure 4a shows time-resolved 2D $^{13}$C ssNMR spectra with $\tau_{sd} = 20$ ms for samples in which Aβ40 was $^{13}$C-labeled at all carbon sites of V18, A30, and G33 (Aβ40-VAG). The full set of 2D spectra and representative 1D slices are shown in Supplementary Fig. 5. In this case, the smaller number of labeled residues allows individual crosspeaks to be resolved. With the higher resolution, differences in crosspeak shapes between samples with $\tau_e = 1.5$ ms, 400 ms, and 1.0 h are visible, consistent with a progressive increase in conformational order. Significant changes in $^{13}$C chemical shifts from crosspeak positions at $\tau_e = 0$ to those at $\tau_e \geq 1.5$ ms are also apparent. Heat maps of pairwise rmsd values in Fig. 4b, c show that differences between the 2D spectrum of Aβ40-VAG with $\tau_e = 0$ and 2D spectra with $\tau_e \geq 1.5$ ms (rmsd values of 0.77 ± 0.10 and 0.82 ± 0.13 in Fig. 4b, c, respectively) are greater than differences among 2D spectra with $\tau_e \geq 1.5$ ms (rmsd values of 0.55 ± 0.05 and 0.55 ± 0.08 in Fig. 4b, c, respectively).

Partial $^{13}$C chemical shift assignments from the 2D spectra of frozen solutions containing Aβ40 monomers ($\tau_e = 0$, pH 12), oligomers ($\tau_e > 0$), and fibrils are compared in Table 1. Chemical shifts in

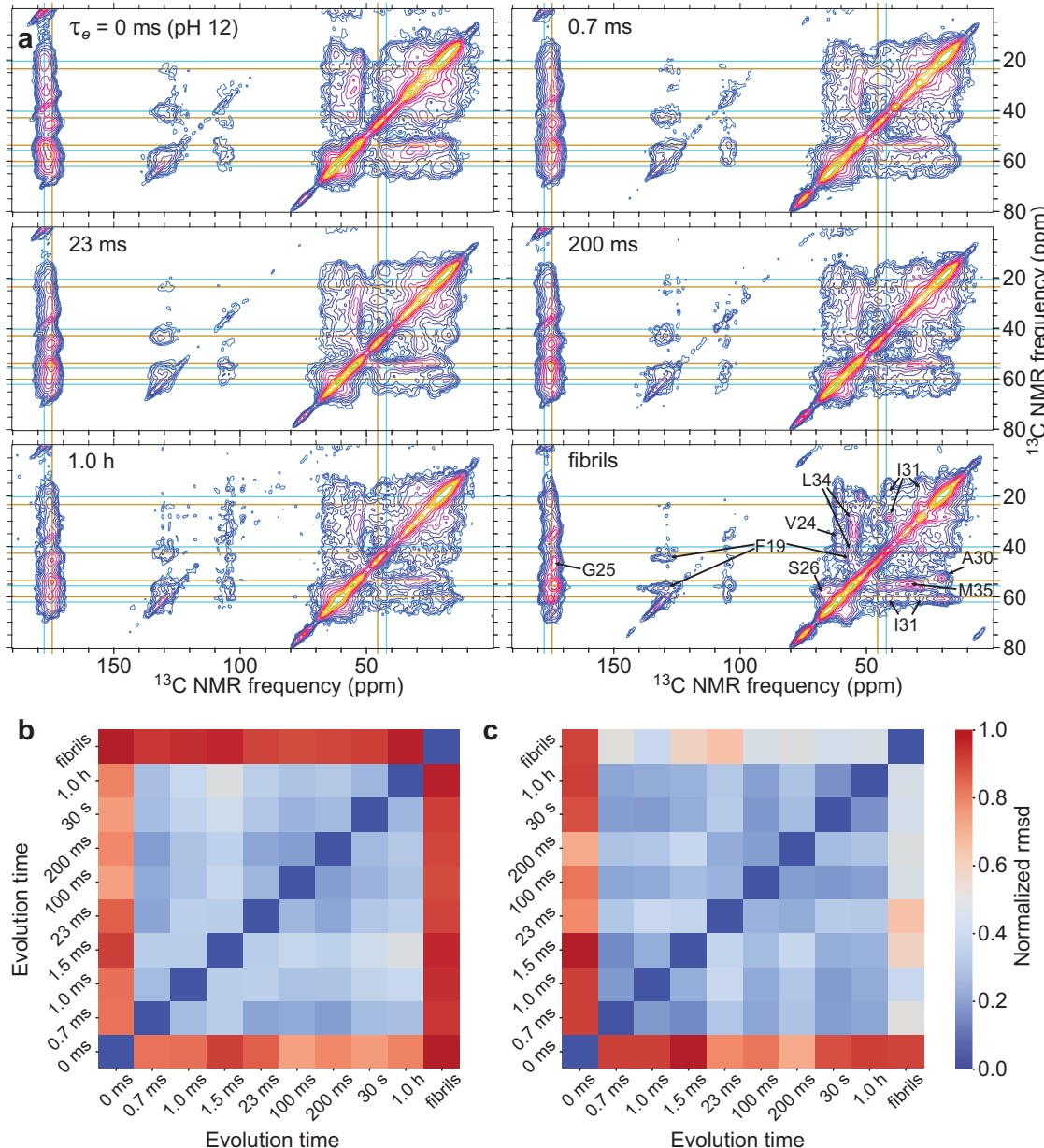

**Fig. 3 | Time-resolved 2D ssNMR spectra of Aβ40 assemblies. a** 2D $^{13}$C-$^{13}$C ssNMR spectra of frozen Aβ40-FVGSAILM solutions with the indicated values of the evolution time $\tau_e$. 2D spectra were recorded with 20 ms mixing periods, sufficient to produce strong intra-residue crosspeaks but not inter-residue crosspeaks. Horizontal and vertical lines indicate positions of crosspeak signal maxima that differ between spectra at $\tau_e = 0$ (cyan lines) and $\tau_e > 0$ (gold lines). Residue-specific assignments of crosspeaks are shown in the 2D spectrum of Aβ40-FVGSAILM fibrils, where the crosspeaks are sharper due to greater structural order. Contour levels increase by factors of 1.3. **b** Heat map plot of differences in crosspeak intensity patterns, quantified by rmsd values, for all pairs of 2D ssNMR spectra. Values are normalized to the maximum rmsd. Only off-diagonal intensities in the aliphatic-aliphatic regions of the 2D spectra are included. **c** Same as panel **b**, but for carbonyl-aliphatic regions of the 2D spectra. Source data are provided as a Source data file.

this table represent values at the maxima of resolved or partially resolved crosspeaks. Full-width-at-half-maximum (FWHM) linewidths were estimated from the crosspeak shapes where possible. The upfield shifts of $^{13}$CO and/or $^{13}$C$_\alpha$ signals of V18, F19, V24, A30, I31, G33, and M35 by more than 1.0 ppm in 2D spectra of Aβ40 oligomers, relative to the 2D spectrum of monomers, indicate the development of a preference for β-strand conformations at these residues. Downfield shifts by more than 1.0 ppm for $^{13}$C$_\beta$ signals of V18, F19, A30, I31, and L34 also indicate the development of β-strand conformations. The relatively small (for the Aβ40-VAG labeling pattern) or undetectable (for the Aβ40-FVGSAILM labeling pattern) differences between 2D spectra with the shortest non-zero $\tau_e$ values and with $\tau_e = 1.0$ h indicate that site-specific molecular conformational

distributions do not change greatly after the initial rapid conformational transition.

$^{13}$CO, $^{13}$C$_\alpha$, $^{13}$C$_\beta$ chemical shifts of labeled residues in the monomeric state are within 1.0 ppm of random coil values[45], with the exceptions of $^{13}$C$_\alpha$ of V24, $^{13}$C$_\alpha$ and $^{13}$C$_\beta$ of A30, $^{13}$C$_\alpha$ of I31, and $^{13}$CO and $^{13}$C$_\alpha$ of L34. For V24, A30, and L34, the differences from random coil values are not consistent with β-strand conformations.

**Evolution of oligomer sizes from time-resolved light scattering**
The time-resolved ssNMR data show that Aβ40 molecules undergo large changes in secondary structure preferences within 0.7–1.5 ms after a rapid change from solvent conditions that favor the monomeric state to conditions that favor self-assembly. However, the time-

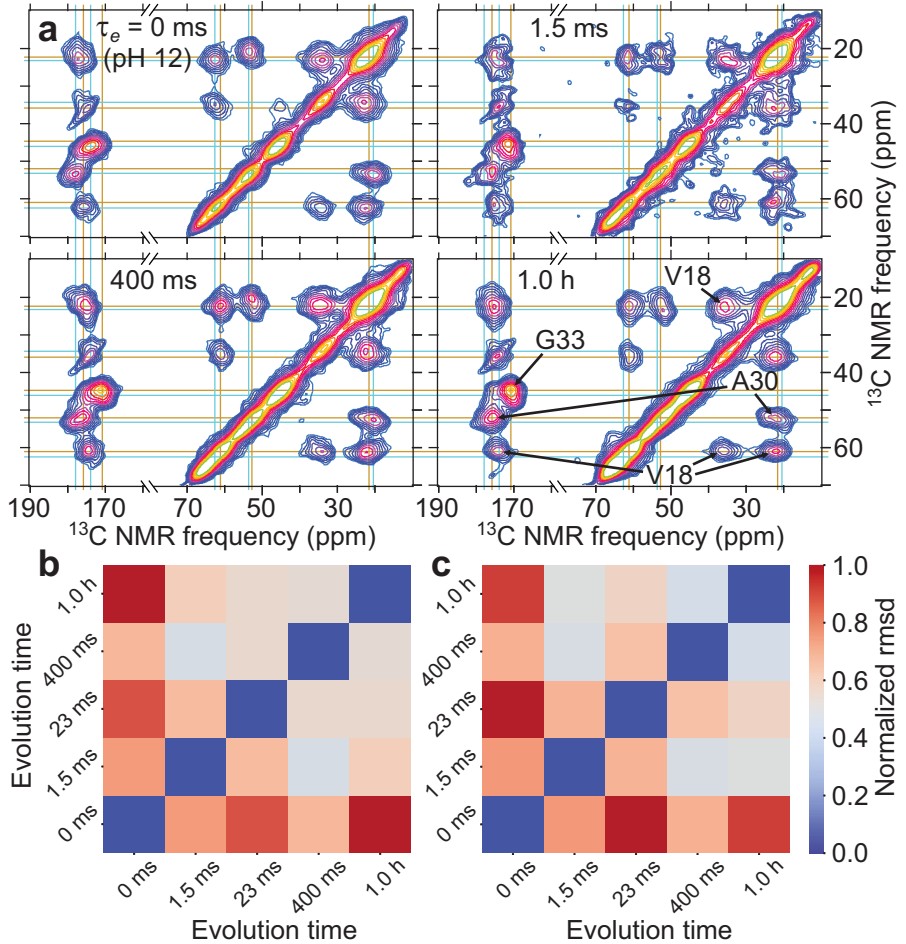

**Fig. 4 | Additional time-resolved 2D ssNMR spectra of Aβ40 assemblies. a** 2D $^{13}$C-$^{13}$C ssNMR spectra of frozen Aβ40-VAG solutions with the indicated values of the evolution time $\tau_e$. 2D spectra were recorded with 20 ms mixing periods. Horizontal and vertical lines indicate positions of crosspeak signal maxima that differ between spectra at $\tau_e = 0$ (cyan lines) and $\tau_e > 0$ (gold lines). Contour levels increase by factors of 1.2. **b** Heat map plot of differences in crosspeak intensity patterns, quantified by rmsd values, for all pairs of 2D ssNMR spectra. Values are normalized to the maximum rmsd. Only off-diagonal intensities in the aliphatic-aliphatic regions of the 2D spectra are included. **c** Same as **b**, but for carbonyl-aliphatic regions of the 2D spectra. Source data are provided as a Source data file.

dependent size of Aβ40 assemblies can not be determined from these data. Thus, from the ssNMR data alone, it is unclear whether the development of β-strand secondary structure depends on the formation of large assemblies or how these assemblies change in size over the time range probed by the ssNMR data.

To characterize the time-dependent sizes of Aβ40 assemblies, we used a stopped flow fluorescence instrument to perform time-resolved light scattering measurements, setting the detection wavelength equal to the excitation wavelength (see "Methods" section). The compositions of the two solutions that were rapidly mixed to initiate Aβ40 self-assembly in these stopped flow measurements were identical to those in the time-resolved ssNMR measurements. For a solution of homogeneous molecular species with molecular weight $M_w$ and mass concentration $c$, light scattering signal intensities, measured as voltages from a photomultiplier tube (PMT) detector, are expected to be proportional to $S_b + c \times M_w$, where $S_b$ is a constant background level from the solvent[46,47]. Measurements with the stopped flow instrument on proteins with various values of $M_w$ verify this expectation (see Supplementary Fig. 6). For measurements on Aβ40 solutions that contain $n$-mers with mass concentrations $c_n(t)$ at time $t$, the light scattering signal is then proportional to $S(t) = S_b + M_w \sum_{n=1}^{\infty} [c_n(t) \times n]$, with $M_w = 4.33$ kDa being the molecular weight of Aβ40 monomers. If monomers at $t = 0$ were to convert completely to octamers at $t = \infty$, for example, $S(t) - S_b$ would increase by a factor of eight, since in that case

$c_8(\infty) = c_1(0)$. In general $S(t) - S_b$ is proportional to the mass-weighted average value of $n$, defined by $n_{ave}(t) = \sum_{n=1}^{\infty} [c_n(t) \times n] / \sum_{n'=1}^{\infty} c_{n'}(t)$.

Figure 5a, b show time-resolved light scattering data for Aβ40, acquired with the highest accessible time resolution of the instrument (0.25 ms time steps). At 1.5 mM and pH 12, Aβ40 monomers produce a scattering signal that is 0.015 V above the buffer scattering level. After a rapid pH drop, the scattering signal rises with a time dependence that can be fit with the stretched-exponential expression $S(t) - S_b = A_1 + B_1\{1 - \exp[-(t/\tau_1)^{\beta_1}]\}$ with $A_1 = 0.015$ V, $B_1 = 0.1147 \pm 0.0033$ V, $\tau_1 = 141 \pm 14$ ms, and $\beta_1 = 0.540 \pm 0.020$. Thus, on the time scale of 0.5 s, Aβ40 monomers self-assemble to form oligomers with $n_{ave} = B_1/A_1 \approx 8$. Importantly, the time required for the light scattering signal above background to double is approximately 10 ms (Fig. 5a inset). Combined with the time-resolved ssNMR results, which show changes in $^{13}$C chemical shifts with 0.7 ms ≤ $\tau_e$ ≤ 1.5 ms, the light scattering data indicate that Aβ40 molecules develop β-strand secondary structure in their monomeric state after a rapid pH drop.

A 2D ssNMR spectrum of Aβ40-FVGSAILM in frozen solution with [Aβ40] = 0.35 mM and $\tau_e = 0.7$ s is nearly identical to the corresponding 2D spectrum with [Aβ40] = 1.5 mM (Supplementary Fig. S7), providing further support for the development of β-strand secondary structure in the monomeric state of Aβ40 after a rapid change to solvent conditions that favor self-assembly. Over longer time periods, light scattering signals continue to grow (Fig. 5b), indicating $n_{ave} \approx 50$ at $t = 600$ s

**Table 1 | ¹³C ssNMR chemical shifts and linewidths in frozen solutions of Aβ40 in monomeric (mono), oligomeric (oligo), and fibrillar (fib) states, determined from 2D ¹³C-¹³C ssNMR spectra in Figs. 3 and 4**

| Residue | CO | | | | Cα | | | | Cβ | | | | Cγ | | | |
|---|---|---|---|---|---|---|---|---|---|---|---|---|---|---|---|---|
| | mono | oligo | fib | coil | mono | oligo | fib | coil | mono | oligo | fib | coil | mono | oligo | fib | coil |
| V18 | 175.6±2.1 | 174.2±2.0 | nd | 176.3 | 62.5±2.5 | 61.0±1.8 | nd | 62.2 | 34.5±2.5 | 36.0±2.0 | nd | 32.9 | 22.4±2.7 | 22.0±2.9 | nd | 21.1, 20.3 |
| F19 | ur | ur | 173.4±1.2 | 175.8 | 58.5* | 57.0±2.0 | 56.3±1.8 | 57.5 | 40.3±3.0 | 42.6±3.0 | 44.3±1.3 | 39.6 | | | | |
| V24 | 176.1±2.7 | ur | 174.8±1.5 | 176.3 | 63.3±3.0 | 61.7±2.5 | 60.8±1.3 | 62.2 | 32.0* | 33.0±3.0 | 35.7* | 32.9 | ur | ur | 21.3* | 21.1, 20.3 |
| G25 | 175.5±2.7 | 174.7±2.4 | 173.6±1.6 | 174.9 | 45.7±1.9 | 45.5±2.5 | 47.0±2.1 | 45.1 | | | | | | | | |
| S26 | 174.3±1.6 | 174.3±2.0 | 174.5±2.7 | 174.6 | 58.6±2.7 | 58.5±2.0 | 58.0±2.0 | 58.3 | 64.0±2.4 | 64.0±3.0 | 66.2±2.4 | 63.8 | | | | |
| A30 | 177.7±2.2 | 175.6±2.0 | 175.3±1.6 | 177.8 | 53.5±1.5 | 52.1±1.8 | 52.3±1.0 | 52.5 | 20.5±2.5 | 22.2±3.4 | 20.2±1.4 | 19.1 | | | | |
| I31 | 176.1±2.7 | 175.6±2.0 | 174.8±1.5 | 176.4 | 62.3* | 60.6±2.3 | 60.8±1.2 | 61.1 | 39.0±2.8 | 42.3±2.5 | 41.2±1.5 | 38.8 | ur | 28.6±3.6, 17.4±3.2 | 28.6±1.6,18.8* | 27.2, 17.4 |
| G33 | 173.9±2.6 | 170.7±1.6 | nd | 174.9 | 46.1±1.8 | 44.9±1.9 | nd | 45.1 | | | | | | | | |
| L34 | 176.2±2.5 | ur | 173.4±1.5 | 177.6 | 56.2±2.8 | ur | 56.2±1.9 | 55.1 | 42.0±3.4 | 45.5±3.5 | 41.4±1.8 | 42.4 | 27.5±2.9 | 28.8±3.0 | 28.6±1.2 | 26.9 |
| M35 | 177.3±2.3 | 174.7±2.9 | 173.4±1.5 | 176.3 | 55.2±2.8 | 54.6±1.5 | 55.0±1.2 | 55.4 | ur | ur | 36.1* | 32.9 | ur | ur | 33.0* | 32.0 |

Values are listed as s±w, where s is the chemical shift (ppm relative to DSS) and 2w is the approximate FWHM linewidth. Random coil values (coil) are taken from Wishart et al.[45]. nd data not recorded, ur unresolved, precluding chemical shift measurement, * measurement of FWHM precluded by spectral overlap.

and $n_{ave} \approx 150$ at $t = 4000$ s. Remarkably, as discussed above, the time-resolved ssNMR spectra indicate only minor changes in molecular conformational distributions as oligomer sizes increase to these levels.

Our interpretation of the light scattering data is simplistic in that we ignore possible variations of the refractive index increment with oligomer size, effects of inter-particle interactions (i.e., the second virial coefficient), and effects of particle shape[47,48]. Given that TEM images indicate predominantly globular particles that are much smaller than the 562 nm wavelength of light in our experiments (Supplementary Fig. 1a–d) and given that we do not attempt to extract structural information from the data other than the approximate value of $n_{ave}$, this simplistic treatment is justified. To be specific, for randomly oriented spheroidal particles with 524 nm³ volume (10 nm diameter if spherical), the scattering intensity perpendicular to the incident light beam is calculated[47] to vary by only 3% as the aspect ratio of the particles varies between 0.3 (oblate) and 3.0 (prolate).

## Modeling of oligomer growth as a coagulation process

A striking feature of the data in Fig. 5b is the nearly linear increase in scattering signal beyond $t = 300$ s. In an attempt to explain this behavior, we considered a simple model for oligomer growth in which oligomers of size $n$ and $m$ can fuse irreversibly to form oligomers of size $n + m$, with rate constants $r_{n,m}$. Such a model describes a process that can be called coagulation[49–52]. In this model, mass concentrations evolve with time according to the equations

$$\frac{dc_n(t)}{dt} = \begin{cases} -\sum_{m=1}^{\infty} \frac{r_{n,m}c_n(t)c_m(t)}{m}(1+\delta_{n,m}), n=1 \\ \sum_{m=1}^{n/2} \frac{nr_{m,n-m}c_m(t)c_{n-m}(t)}{m(n-m)} - \sum_{m=1}^{\infty} \frac{r_{n,m}c_n(t)c_m(t)}{m}(1+\delta_{n,m}), n=2,4,6,\ldots \\ \sum_{m=1}^{(n-1)/2} \frac{nr_{m,n-m}c_m(t)c_{n-m}(t)}{m(n-m)} - \sum_{m=1}^{\infty} \frac{r_{n,m}c_n(t)c_m(t)}{m}(1+\delta_{n,m}), n=3,5,7,\ldots \end{cases}$$

(1)

Importantly, Eq. (1) conserve total mass, i.e., $\sum_{n=1}^{\infty} \frac{dc_n(t)}{dt} = 0$.

If rates of oligomer fusion were purely diffusion-limited, and if oligomers were approximately spherical with radii $R_n$ and translational diffusion constants $D_n$, then $r_{n,m} \approx 4\pi(D_n + D_m)(R_n + R_m)$[49,50]. Based on the Stokes–Einstein equation $D_n = k_BT/(6\pi\eta R_n)$, where $k_B$ is the Boltzmann constant and $\eta$ is the solvent viscosity, and the relation $R_n \propto n^{1/3}$, we therefore assume that $r_{n,m} = (2 + \frac{m^{1/3}}{n^{1/3}} + \frac{n^{1/3}}{m^{1/3}}) \times r_0$, where $r_0$ is an overall scaling factor for the oligomer fusion rates. Numerical solutions of Eq. (1) with this simple expression for $r_{n,m}$ show nearly linear dependences of the simulated light scattering signals on time (Supplementary Fig. 8a), in agreement with the long-time behavior of the experimental data. We note that closely related treatments of coagulation processes have been described previously[49–52].

To reproduce the rapid, nonlinear time dependence of experimental light scattering signals at shorter times, we introduce a rate enhancement function $E(n,m)$, so that $r_{n,m} = E(n,m) \times (2 + \frac{m^{1/3}}{n^{1/3}} + \frac{n^{1/3}}{m^{1/3}}) \times r_0$. Since the experimental data imply that fusion rates are relatively large when the oligomers are small, we assume $E(n,m) = 1 + (E_0 - 1) \exp[-(n^2 + m^2)/N_{th}^2]$. With this form for $E(n,m)$, fusion rates are enhanced by approximately $E_0$ when $\sqrt{n^2 + m^2}$ is less than or comparable to a threshold value $N_{th}$.

Figure 5c compares the experimental light scattering data at [Aβ40] = 1.5 mM with simulated data for various values of $N_{th}$. In these plots, light scattering signals are normalized to the signal from a 1.0 mM solution of Aβ40 monomers and background scattering is subtracted. Values of $r_0$ and $E_0$ were optimized at each value of $N_{th}$ by minimizing the squared deviation $s^2$ between simulated and experimental data. To simplify the $s^2$ calculations, experimental data were represented by an empirical function of the form $S(t) - S_b = A_1 + A_2t + B_1\{1 - \exp[-(t/\tau_1)^{\beta_1}]\} + B_2\{1 - \exp[-(t/\tau_2)^{\beta_2}]\}$, using values of $A_1$,

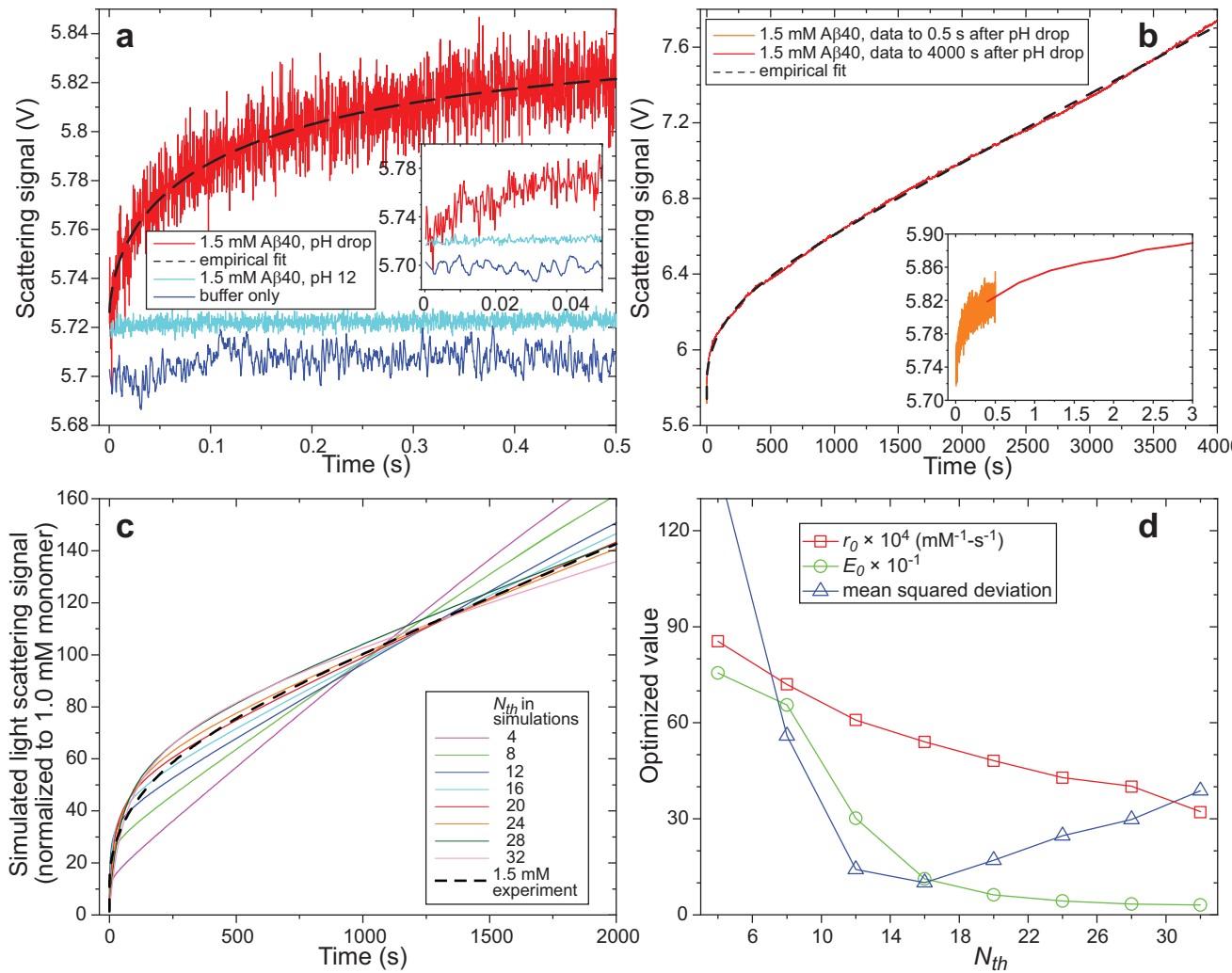

**Fig. 5 | Quantification of Aβ40 oligomer sizes by time-resolved light scattering.**
**a** Light scattering signals, measured as photomultiplier tube voltages, for a
1.5 mM Aβ40 solution at pH 12 (cyan), a 1.5 mM Aβ40 solution after a rapid pH drop
from 12 to 7.4 (red), and a pH 7.4 buffer alone (blue). Dashed line is a stretched-
exponential fit to the pH drop data, as described in the text. Inset shows the data up
to 50 ms. **b** Light scattering data recorded to 3600 s after a rapid pH drop. Insets
compare the data up to 3.0 s (red) with pH drop data from **a** (orange). Dashed line
is an empirical fit to a function that includes two stretched-exponential terms to
describe curvature on 100 ms and 100 s time scales and a linear term to describe
the long-time behavior, as described in the text. **c** Fits of the experimental data

(dashed line) with simulations based on the coagulation model described in the
text. Simulations parameters $r_0$ and $E_0$ were optimized for each value of the
threshold size $N_{th}$, below which oligomer fusion rates $r_0$ are multiplied by the
enhancement factor of $E_0$. Experimental and simulated light scattering signals are
normalized to the signal from a 1.0 mM solution of monomeric Aβ40.
**d** Dependences of the optimized values of $r_0$ and $E_0$ and the deviation between
optimized simulations and experimental data on the assumed oligomer threshold
size $N_{th}$. The best fit is obtained with $N_{th} \approx 16$. Source data are provided as a Source
data file.

$B_1$, $\tau_1$, and $\beta_1$ determined from data with $t \leq 0.5$ s as described above and
adjusting $A_2$, $B_2$, $\tau_2$, and $\beta_2$ to fit the data. Best-fit values (resulting in the
dashed line in Fig. 5b) were $A_2 = 0.00036057 \pm 0.00000027$ V/s,
$B_2 = 0.43465 \pm 0.00088$ V, $\tau_2 = 176.0 \pm 1.2$ s, and $\beta_2 = 0.6127 \pm 0.0029$.

Within the context of this simple model, the best agreement
between simulated and experimental light scattering data at [Aβ40] =
1.5 mM is achieved with $N_{th} \approx 16$, $r_0 \approx 0.0054$ mM$^{-1}$s$^{-1}$, and $E_0 \approx 120$, as
shown in Fig. 5d. Simulated time dependences of individual oligomer
concentrations with these parameters are shown in Supplementary
Fig. 8b. Although agreement with experimental data is not fully
quantitative, the simulations reproduce the shape and amplitude of
the data over the full time range examined in the experiments.

If oligomer fusion were indeed diffusion limited, we would expect
$r_0 \approx \frac{2}{3} k_B T / \eta = 8.3 \times 10^5$ mM$^{-1}$s$^{-1}$, with $T = 297$ K and $\eta = 2.0$ cP for our
glycerol/water solutions. That the best-fit values of $r_0$ are much smaller
than the diffusion limited value, even when the best-fit enhancements
$E_0$ are included, indicates that Aβ40 oligomer growth is far from being

diffusion limited in our experiments, even for small oligomers.
Apparently, oligomer fusion occurs only rarely when oligomers collide
with one another. This conclusion seems consistent with TEM images,
which show clusters of oligomers with various sizes, in contact with
one another after adsorption and drying on the TEM grid but not fused
(Supplementary Fig. 1a–d).

Time-resolved light scattering data were also acquired at [Aβ40] =
0.75 mM and analyzed with the same approach (Supplementary
Fig. 8c–f). Equation (1) predicts that a twofold reduction in the initial
monomer concentration will simply retard the evolution to oligomers
by a factor of two (because these equations are invariant to the sub-
stitutions $c_n(t) \to x c_n(t)$ and $t \to t/x$ for all n and any x). Although this
prediction is approximately confirmed, in that the scattering signal
above background at $t = 300$ s for [Aβ40] = 1.5 mM is 2.3 times greater
than the signal above background at $t = 600$ s for [Aβ40] = 0.75 mM,
the best-fit functional forms and best-fit values of $r_0$, $E_0$, and $N_{th}$ are
somewhat different at the two concentrations. Given the simplicity of

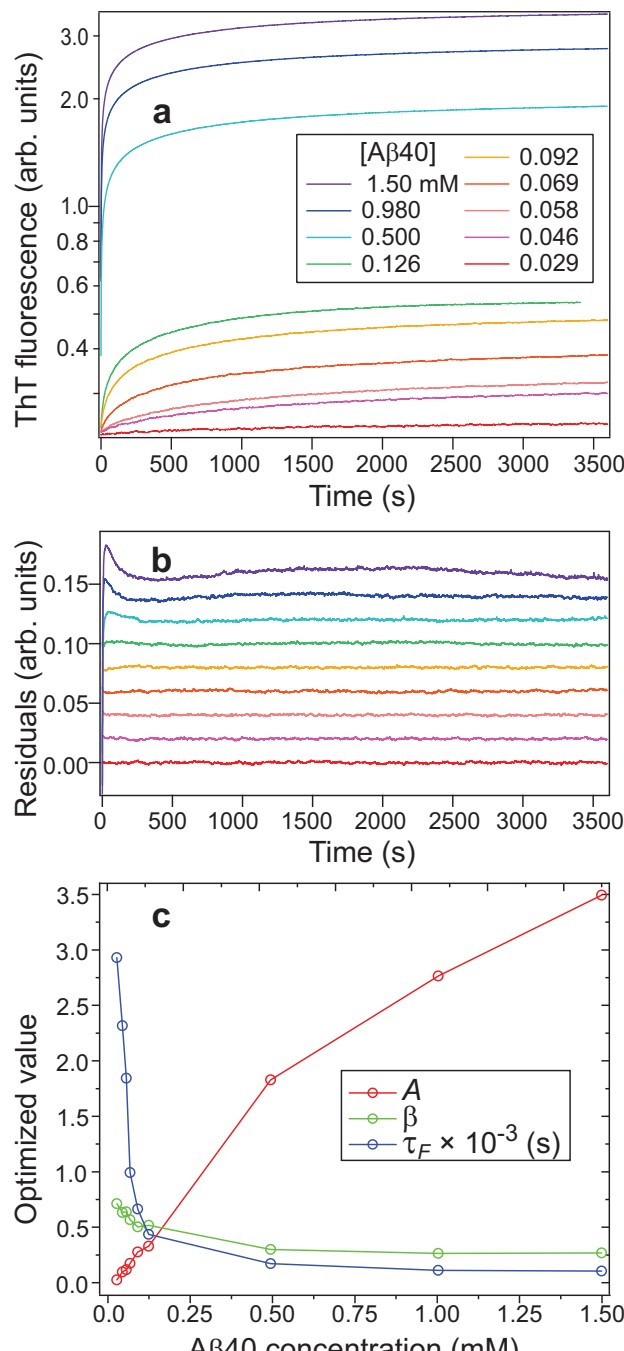

**Fig. 6 | Time dependence of ThT fluorescence from Aβ40 assemblies. a** Stopped-flow ThT fluorescence data for the indicated Aβ40 concentrations after a rapid pH drop. **b** Residuals after fitting the ThT fluorescence data with the stretched exponential function $F(t) = F_0 + A\{1 - \exp[-(t/\tau_F)^\beta]\}$, with $F_0 = 0.23$ representing fluorescence from unbound ThT. For clarity, residuals at increasing color-coded Aβ40 concentrations are offset vertically in increments of 0.02. **c** Best-fit values of the fitting parameters. Uncertainties are smaller than the symbols. Source data are provided as a Source data file.

the coagulation model embodied in Eq. (1) and the form of $r_{n,m}$ used in simulations, it is not surprising that discrepancies exist.

## Evolution of thioflavin T fluorescence intensity

Thioflavin T (ThT) fluorescence is commonly used to assess fibril formation by Aβ and other amyloidogenic polypeptides, as the fluorescence quantum yield increases greatly when ThT becomes

conformationally constrained upon binding to amyloid fibrils[53]. ThT fluorescence upon binding to oligomers has also been reported[5,11,54]. Fig. 6a shows data from stopped flow fluorescence experiments in which Aβ40 solutions at pH 12 were rapidly mixed with concentrated pH 7.4 buffer solutions containing 50 μM ThT, producing final Aβ40 concentrations from 29 μM to 1.5 mM and 25 μM ThT. Fluorescence intensities $F(t)$ increase with characteristic build-up times $\tau_F$ in the 100–500 s range for [Aβ40] > 0.1 mM, as determined by fits with stretched exponential functions of the form $F(t) = F_0 + A\{1 - \exp[-(t/\tau_F)^\beta]\}$ (Fig. 6b, c). Best-fit values of $A$ and $\tau_F$ are approximately proportional to and inversely proportional to the initial Aβ40 monomer concentration, respectively.

In contrast to the time-resolved light scattering signals, ThT fluorescence intensities do not increase linearly at long times. Instead, the combined light scattering and fluorescence data at [Aβ40] = 1.5 mM indicate that the fluorescence signal per Aβ40 molecule increases with oligomer size until $n_{ave} \approx 70$, after which the fluorescence signal per molecule becomes nearly constant while $n_{ave}$ continues to increase linearly. If ThT fluorescence intensity is a signature of β-sheet structure, as is commonly assumed, then these data suggest an increase in the fraction of molecules that participate in β-sheets within nonfibrillar assemblies up to $n_{ave} \approx 70$, but relatively little change as $n_{ave}$ increases further. A spherical assembly containing 70 Aβ40 molecules would have a diameter of approximately 10 nm.

## Evolution of inter-residue contacts from time-resolved ssNMR

2D $^{13}C$-$^{13}C$ ssNMR spectra obtained with longer spin diffusion mixing periods ($\tau_{sd} = 1.0$ s) exhibit crosspeaks between signals from different $^{13}C$-labeled residues when the inter-residue $^{13}C$-$^{13}C$ distances are roughly 6–8 Å or less[6,7,17,18,34,38,39]. Fig. 7a shows such 2D spectra of Aβ40-FVGSAILM samples with several $\tau_e$ values. The full set of 2D spectra is shown in Supplementary Fig. 9. At $\tau_e = 0.7$ ms and $\tau_e = 23$ ms, strong crosspeak intensity that connects $^{13}C$ chemical shifts of the F19 aromatic sidechain near 132 ppm with $^{13}C$ chemical shifts of aliphatic sidechains in the 15–35 ppm range. Crosspeak intensity in this region is significantly weaker at $\tau_e = 0$. As shown in Fig. 7b, the inter-residue aromatic/aliphatic crosspeak volume, relative to the intra-residue F19 $C_\beta$/aromatic crosspeak volume, is independent of $\tau_e$ from 0.7 ms to 1.0 h. Residues that could contribute to the inter-residue crosspeak volume include V24, A30, I31, L34, and M35.

The broad, overlapping lineshapes in these 2D spectra prevent unambiguous assignment of aromatic/aliphatic crosspeak intensity to specific residues. However, in light of the evidence from ssNMR for β-strand secondary structure at V18-V24 and A30-M35 discussed above and the evidence from time-resolved light scattering measurements for a primarily monomeric state in samples with 0.7 ms ≤ $\tau_e$ ≤ 1.5 ms, a reasonable interpretation of the results in Fig. 7a, b is that the Aβ40 conformational distribution favors U-shaped or hairpin-like conformations that bring the F19 sidechain in proximity with sidechains of L34 and/or M35 after the pH drop. With this interpretation, the aromatic/aliphatic crosspeak volume arises from intramolecular contacts. Such conformations in Aβ40 monomers and small oligomers may resemble the U-shaped conformations in ssNMR-based structural models for protofibrillar and fibrillar Aβ40 assemblies[10,17,19,30], or the β-hairpins observed in molecular dynamics simulations[42] and in some structural studies[26,36]. As oligomers grow, the possibility exists that intramolecular aromatic/aliphatic contacts could be replaced to some extent by intermolecular contacts.

Figure 7c shows 2D spectra of Aβ40-VAG samples with $\tau_{sd} = 1.0$ s and various values of $\tau_e$. With this labeling pattern, we observe inter-residue crosspeaks that connect the $^{13}C_\alpha$ chemical shift of G33 (45 ppm) with the $^{13}C_\alpha$ and $^{13}C_\gamma$ chemical shifts of V18 (61 ppm and 22 ppm, respectively). As shown in Fig. 7d, the inter-residue crosspeak volumes, relative to intra-residue crosspeak volumes of V18, are nearly unchanged from $\tau_e = 0$ to $\tau_e = 1.5$ ms but are larger at $\tau_e = 400$ ms and

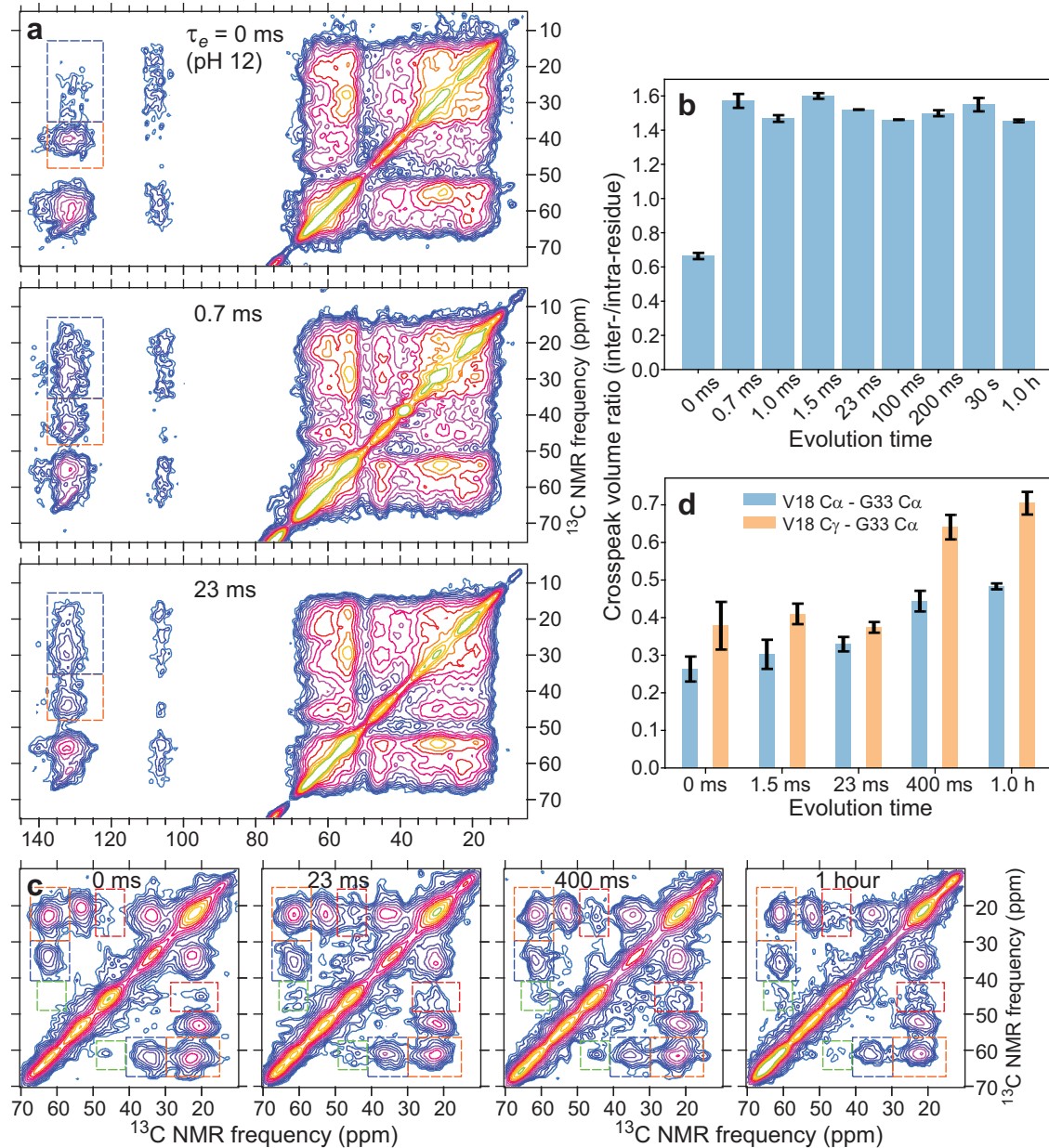

**Fig. 7 | Time dependence of inter-residue contacts in Aβ40 assemblies. a** 2D $^{13}$C-$^{13}$C ssNMR spectra of frozen Aβ40-FVGSAILM solutions with the indicated values of the evolution time $\tau_e$, recorded with 1.0 s mixing periods for detection of inter-residue crosspeaks. Dashed orange and blue rectangles enclose regions of F19 intra-residue crosspeak and F19-V24/A30/I31/L34/M35 inter-residue crosspeak intensity, respectively. Contour levels increase by factors of 1.2. **b** Dependence on $\tau_e$ of the ratio of F19-V24/A30/I31/L34/M35 inter-residue crosspeak volume to F19 intra-residue crosspeak volume. **c** 2D $^{13}$C-$^{13}$C ssNMR spectra of frozen Aβ40-VAG solutions, recorded with 1.0 s mixing periods. Dashed orange and blue rectangles enclose regions of V18 $C_\alpha$-$C_\gamma$ and $C_\alpha$-$C_\beta$ intra-residue crosspeak intensity, respectively. Dashed green and red rectangles enclose regions of V18 $C_\alpha$-G33 $C_\alpha$ and V18 $C_\gamma$-G33 $C_\alpha$ inter-residue crosspeak intensity, respectively. Contour levels increase by factors of 1.3. **d** Dependences on $\tau_e$ of the ratios of V18 $C_\alpha$-G33 $C_\alpha$ and V18 $C_\gamma$-G33 $C_\alpha$ inter-residue crosspeak volumes to V18 $C_\alpha$-$C_\beta$ and $C_\alpha$-$C_\gamma$ intra-residue crosspeak volumes, respectively. In panels b and d, each data point comes from one 2D spectrum (*n* = 1). Error bars are uncertainties calculated from the root-mean-squared noise values in the 2D spectra.

$\tau_e$ = 1.0 h. This behavior is clearly different from the behavior of aromatic/aliphatic crosspeaks involving F19 discussed above. We interpret the increase in V18-G33 crosspeak volumes as the result of an increasing fraction of Aβ40 molecules that participate in intermolecular contacts.

In the previously characterized in-register parallel β-sheet structures of Aβ40 fibrils[17,18,25,26] and the antiparallel β-sheet structure of Iowa-mutant Aβ40 protofibrils[10], the shortest intermolecular or intramolecular V18-G33 distances are 10 Å or more. The observation of strong V18-G33 crosspeaks suggests that neither type of β-sheet is the predominant mode of intermolecular association in nonfibrillar oligomers. Instead, Aβ40 molecules pack in alternative configurations that create closer V18-G33 contacts. One possibility is intermolecular hydrogen bonding between molecules with hairpin-like conformations, for example as suggested recently for the partially disordered outer layers of a brain-derived Aβ40 fibril polymorph[26].

## Discussion

1D and 2D ssNMR spectra of rapidly frozen Aβ40 solutions show obvious differences in $^{13}$C chemical shifts between pH 12 solutions and solutions that were frozen at evolution times $\tau_e$ < 2 ms after a rapid change to pH 7.4 (Figs. 2b, 3a, and 4a). $^{13}$C chemical shifts at pH 12 are

consistent with conformational distributions that lack regular secondary structure, while $^{13}C$ chemical shifts after the rapid pH drop indicate a preference for β-strand secondary structure at residues 18–24 and 30–35 (Table 1). Time-resolved light scattering measurements (Fig. 5a) show that Aβ40 is still primarily monomeric at $\tau_e = 2$ ms. Therefore, although the conformational distribution of Aβ40 monomers does not include a high population of β-strand conformations at pH 12, β-strand conformations become highly populated in Aβ40 monomers at pH 7.4 before oligomerization occurs. Moreover, contacts between the aromatic sidechain of F19 and aliphatic sidechains in residues 30–35 are already present at $\tau_e = 0.7$ ms (Fig. 7a, b), consistent with β-hairpin or strand-loop-strand (U-shaped) conformations in the monomeric state.

The light scattering data show that oligomers with an average size $n_{ave} \approx 8$ develop by $\tau_e = 0.5$ s (Fig. 5a). With [Aβ40] = 1.5 mM, the average oligomer size increases rapidly in the initial period of approximately 100 s, then increases more slowly and linearly up to at least 4000 s, with $n_{ave} \approx 50$ at $\tau_e = 600$ s and $n_{ave} \approx 150$ at $\tau_e = 4000$ s (Fig. 5b). Although the average oligomer size increases substantially, changes in crosspeak patterns in 2D $^{13}C$–$^{13}C$ ssNMR spectra are small, indicating only a minor increase in conformational order and no clear changes in conformational preferences. In the case of Aβ40-FVGSAILM, no significant changes are detected in the 2D spectra from $\tau_e = 0.7$ ms to $\tau_e = 1.0$ h (Fig. 3b, c). This result is partly due to the extensive overlap of crosspeak signals from different isotopically labeled residues, which tends to obscure subtle effects. Subtle changes associated with oligomerization are apparent in 1D spectra (Fig. 2b). In the case of Aβ40-VAG, where spectral overlap is less problematic, 2D crosspeaks at large $\tau_e$ values are sharper than at $\tau_e = 1.5$ ms (Fig. 4), indicating greater molecular conformational order in oligomers with $n_{ave} \geq 8$ than in the monomeric state that develops quickly after the pH drop. Moreover, V18-G33 crosspeak volumes increase with $\tau_e$ in 2D spectra with 1.0 s mixing periods (Fig. 7d), consistent with an increasing population of intermolecular contacts that result in relatively short V18-G33 distances.

The linear increase in light scattering intensity (i.e., in $n_{ave}$) for $\tau_e > 500$ s can be fit semi-quantitatively with a simple coagulation model for oligomer growth, in which smaller oligomers fuse irreversibly with one another to generate larger oligomers (Fig. 5c). To reproduce the rapid initial rise in scattering observed experimentally, fusion rates for small oligomers must be enhanced by factors $E_0 \approx 120$ relative to fusion rates for larger oligomers, up to oligomer sizes near a threshold value $N_{th} \approx 16$. This result may reflect increasing sequestration of hydrophobic sidechains within the oligomer core and increasing structural rigidity as the size increases.

Even with large enhancement factors, fusion rates for small oligomers are small compared with theoretical diffusion-limited rates. The observation that $n_{ave} \approx 2$ at $\tau_e = 20$ ms with [Aβ40] = 1.5 mM (Fig. 5a) implies a bimolecular fusion rate for monomers on the order of 40 mM$^{-1}$s$^{-1}$, which is 20,000 times less than the theoretical diffusion-limited rate. The best-fit value $r_0 \approx 0.0054$ mM$^{-1}$s$^{-1}$ for large oligomers (Fig. 5d) is smaller than the diffusion-limited rate by a factor of $10^8$. Thus, the vast majority of collisions do not lead to fusion and growth of oligomers. Apparently, specific sets of intermolecular contacts are required to stabilize the association of two non-fibrillar Aβ40 entities. A similar conclusion was reached in an earlier quantitative study of fibril growth kinetics[55], where the dependence of fibril extension rates on monomer concentration implied that approximately $10^4$ collisions were required to add one monomer to the end of a fibril.

Previous experimental studies provide relatively little information about molecular structural evolution in early Aβ assemblies that can be compared with our time-resolved ssNMR data. Solution NMR experiments by Barnes et al.[56], in which Aβ40 self-assembly was initiated by a rapid drop in pressure from 2.5 kbar to 1 bar, showed that oligomerization occurred in less than 1 s at [Aβ40] = 1.3 mM. Residues 10–40 became at least partially ordered within several seconds, with maximum ordering in residues 16–22 and 29–36, consistent with the β-strand formation seen in our time-resolved ssNMR measurements. Measurements of $^{15}N$ transverse relaxation rates showed that oligomer sizes were roughly 350 kDa after 1.25 s and 1.3 MDa after 3.75 s, corresponding to $n_{ave} \approx 80$ and $n_{ave} \approx 300$, respectively. In TEM images, Aβ40 assemblies generated by the pressure drop were primarily protofibrillar. The relatively rapid development of large assemblies with protofibrillar morphology in the experiments of Barnes et al., as opposed to the slower development of globular assemblies in our experiments, may be due to differences in buffer compositions, differences in the properties of soluble Aβ40 at high pressure or high pH, or differences in the molecular structural evolution that occurs as pressure is released or the pH is reduced.

On longer time scales, Aβ self-assembly processes have been characterized by techniques such as photochemical crosslinking[3], dynamic and static light scattering[48,57,58], and imaging methods[4,8,59]. These measurements provide information about the masses, dimensions, and morphologies of Aβ assemblies but do not provide structural information at the level of molecular conformation or intermolecular interactions.

Based on dynamic light scattering data for Aβ40 solutions at low pH acquired on time scales of 25–50 h, Lomakin et al. proposed a mechanism for Aβ40 fibril formation in which monomers form micelle-like assemblies above a critical concentration, nucleation of fibrils occurs within micelles, and fibrils grow by monomer addition[58,60]. This work is not directly related to our results, although the coagulation model used to fit our light scattering data up to 1.0 h may be considered a mechanism for formation of micelle-like assemblies. The existence of a critical concentration above which the process of fibril formation includes non-fibrillar assemblies as intermediates is consistent with the experiments of Hasecke et al. on self-assembly of a dimeric Aβ40 construct[11], in which ThT fluorescence build-up curves were found to be biphasic above a threshold concentration, concomitantly with the appearance of globular and protofibrillar assemblies in atomic force microscope images. The data of Hasecke et al. support a picture in which larger globular and protofibrillar assemblies are metastable off-pathway intermediates[11]. The off-pathway nature of large globular and protofibrillar assemblies is also implied by ssNMR measurements on protofibrils[10,34,36] and globular oligomers[6,7,37], which show that the β-sheet structures in these assemblies differ qualitatively from the in-register parallel β-sheets in mature Aβ fibrils.

Pallitto and Murphy proposed a rather different mechanism for Aβ40 fibril formation, also based on dynamic and static light scattering data acquired on a 40 h time scale[48,57]. Their mechanism does not include large non-fibrillar assemblies and therefore does not apply to our results.

More recently, Knowles and coworkers proposed a model for Aβ self-assembly that includes oligomers as a class of intermediates on the pathway from monomers to fibrils, with variable rates of conversion from oligomeric to monomeric or fibrillar states[13,61]. Structural properties of oligomers were not specified. Fitting of experimental data for time-dependent fibril and oligomer quantities led to a conclusion that Aβ40 and Aβ42 oligomers have similar lifetimes, on the order of several hours, but differ in their probability of converting to a fibrillar state[61]. Although this model explains the experimental results of Knowles and coworkers, the greater Aβ40 concentration and other differences in our experiments make this model inapplicable.

As with previous in vitro studies of aggregation mechanisms, not all results from our time-resolved ssNMR and light scattering experiments are expected to be directly relevant to Aβ self-assembly in vivo, since the peptide concentrations in our experiments are relatively high, the maximum time is relatively short, and other biochemical components and biological processes are not included. Aspects that may be relevant to self-assembly in vivo include the observed preference for β-strand conformations at the monomer level, the

observed difference in oligomer fusion rates for oligomer sizes below and above $n \approx 16$, and the absence of ordered β-sheet structures within the oligomers, even when $n_{ave} > 100$.

Under conditions where a 0.3 mM Aβ40 solution remains primarily monomeric for at least several days (4 °C, pH 7.0, 20 mM sodium phosphate), NMR measurements by Roche et al.[15] indicate an absence of preferred secondary structure. These and other results that support a random coil structure[16] may appear to conflict with our finding of β-strand conformations at $\tau_e = 0.7$–1.5 ms, where the light scattering data indicate a primarily monomeric state. A likely explanation is that the monomer conformation is sensitive to variations in solvent conditions and temperature that modulate the importance of hydrophobic interactions. Conditions that favor self-assembly are those that strengthen intermolecular hydrophobic interactions relative to unfavorable electrostatic interactions and entropic penalties. Such conditions may also promote intramolecular hydrophobic interactions, especially interactions between the hydrophobic side-chains of residues 17–21 (LVFFA sequence) and 30–36 (AIIGLMV sequence) that can stabilize β-hairpin or strand-loop-strand conformations of Aβ40 monomers.

Several groups have reported ssNMR studies of metastable Aβ40[5] and Aβ42[6,7,37] oligomers, prepared with incubation periods greater than 24 h. Results of these studies are generally consistent with our data for early Aβ40 oligomers, with [13]C chemical shifts that support β-strand conformations in residues 17–21 and 31–36, linewidths greater than 2 ppm, crosspeaks from F19 to I31[7] or L34[6] in 2D spectra with long mixing periods, and intermolecular distance constraints that argue against in-register parallel β-sheet organizations similar to those in mature fibrils.

Results described above demonstrate the utility of time-resolved ssNMR as a means of characterizing molecular structural properties of Aβ species that develop in the earliest stages of self-assembly. The combination of time-resolved ssNMR with time-resolved light scattering allows molecular structural properties to be correlated with oligomer size, from several milliseconds to thousands of seconds. The same approach can be applied to other peptide and protein systems that form filaments, capsid shells[62], or other large assemblies, and possibly to phase-separating systems[63]. Self-assembly can be initiated by changes in pH, temperature[64], or ionic strength[62], or by mixing interacting components[39]. Although the broad lines in ssNMR spectra of frozen solutions limit the number of sites that can be isotopically labeled simultaneously, especially for conformationally disordered systems, segmental labeling[65] and heteronuclear filtering[66,67] methods can be used to extend this approach to larger proteins. Thus, we expect the experimental approaches demonstrated above to provide structural and mechanistic insights into a wide variety of biomolecular processes in future studies.

## Methods

### Sample preparation
Aβ40-FVGSAILM was synthesized on a Biotage Initiator+ Alstra solid phase peptide synthesizer and purified by high-performance liquid chromatography (HPLC) with a reverse-phase C3 column. Full details are given in a previous publication[68]. Aβ40-VAG was synthesized and purified with the same methods. [13]C-labeled amino acids, introduced at specific sites by solid phase synthesis, were also [15]N-labeled, but this does not affect the ssNMR results. Lyophilized, HPLC-purified peptides were dissolved initially at 4.6 mM concentration in 40 mM NaOH, then diluted with water/glycerol to produce a final 2.3 mM peptide concentration in 20 mM NaOH, 20% v/v glycerol, pH 12. To initiate self-assembly, Aβ40 solutions at pH 12 were rapidly mixed in a 2:1 ratio with 524 mM sodium phosphate buffer, 20% v/v glycerol/ pH 7.4. For ssNMR measurements, water components were 1:7 $H_2O$:$D_2O$ and the glycerol was perdeuterated and [13]C-depleted (Cambridge Isotope Laboratories). Solutions for ssNMR also contained 10 mM sulfoacetyl-DOTOPA as the paramagnetic dopant for DNP[43]. Solutions were

filtered through 0.45 μm polyvinylidene fluoride centrifugal filters (Millipore) before mixing.

For comparisons between ssNMR spectra of non-fibrillar Aβ40 assemblies with spectra of fibrils, Aβ40-FVGSAILM fibrils were grown by dissolving the peptide at pH 12, mixing with sodium phosphate buffer to prepare a solution with 1.5 mM peptide concentration at pH 7.4 (without glycerol), incubating the solution quiescently at 24 °C for 1.0 h, sonicating the sample for 60 s in a bath sonicator to break fibrils that formed within 1.0 h into fragments that act as seeds, then incubating for an additional 14 h. During the final incubation period, seeds grew into longer fibrils while the less thermodynamically stable non-fibrillar assemblies dissolved, resulting in a sample that was predominantly fibrillar (Supplementary Fig. 1h). Deuterated, [13]C-depleted glycerol and sulfoacetyl-DOTOPA were then added to final concentrations of 40% v/v and 10 mM, respectively. The fibril solution was then loaded into a MAS rotor and frozen by immersion in liquid nitrogen.

### Rapid mixing and freezing for time-resolved ssNMR
Full details of the home-built apparatus for rapid mixing and freezing are given in a previous publication[38]. Briefly, a pair of HPLC pumps drive two solutions through the mixer, which is constructed from a Y-junction with 100 μm inner diameter (ID) channels. The solutions are initially contained in sections of 760 μm ID polyether ether ketone (PEEK) tubing that are connected to the inputs of the Y-junction. In the standard configuration, the output from the Y-junction connects with a 4 mm section of 100 μm ID polyether ether ketone (PEEK) tubing that is packed with 40 μm stainless steel beads (Cospheric LLC) and capped with 10 μm mesh disks (Valco Instruments Co.) Assuming that beads occupy 33% of the available volume, the internal volume of this section is approximately 0.021 μl. The bead-packed section is followed directly by a 10 mm section of 50 μm ID PEEKsil tubing (0.020 μl internal volume) which serves as the output nozzle of the mixer. The mixer is contained in a poly-tetrafluoroethylene compartment maintained at 25 °C and is mounted on a stepper motor driven arm that sweeps across the surface of a rotating copper disk (18 cm diameter, 0.5 cm thickness, spinning at 250 rpm). The copper disk is precooled to 77 K with liquid nitrogen. The mixed solution leaves the nozzle as a high-speed jet (0.85–2.6 cm/ms jet velocity at total flow rates of 1.0–3.0 ml/min), freezing into a glassy state upon striking the cold copper surface. The resulting frozen solution is then packed into a magic-angle spinning (MAS) ssNMR rotor under liquid nitrogen and stored in liquid nitrogen until ssNMR measurements are performed.

We define the structural evolution time $\tau_e$ to be the sum of the average flow time through the mixing section, the average flow time through the nozzle, and the subsequent flight time to the cold copper surface. To achieve the smallest possible values of $\tau_e$, the bead-packed section was omitted and the nozzle length was increased to 14 mm (0.027 μl internal volume). Using nuclear spin relaxation measurements as previously described[38], we verified that complete mixing occurred in this configuration as a result of flow through the mesh disk followed by flow through the nozzle. With the two mixer configurations, mixing times (not including the flight time) in the 0.4–2.5 ms range were achieved with total flow rates between 3.0 ml/min and 1.0 ml/min. With a 0.5 cm flight distance and 3.0 ml/min flow, $\tau_e = 0.7$ ms.

As summarized in Supplementary Table 1, values of $\tau_e$ from 0.7 ms to 1.5 ms were obtained by varying the total flow rate and the distance between the mixer nozzle and the copper surface. Values of $\tau_e$ from 23 ms to 400 ms were obtained by inserting sections of 100 μm or 245 μm ID tubing with appropriate volumes between the mixing section and the nozzle. All values of $\tau_e$ are average values calculated from the flow rates and the IDs and lengths of tubing. Of course, non-uniform flow patterns within the mixer and additional tube sections introduce a range of actual transit times for individual Aβ40 molecules. If fully developed laminar flow existed throughout the system, 67% of the

sample volume collected for ssNMR measurements would have transit times between 0.5 and 1.5 times the nominal values. Deviations from laminar flow reduce the range of transit times. Therefore, we estimate that actual evolution times within each sample vary by approximately ±30% from the nominal $\tau_e$ values.

For $\tau_e$ = 30 s and $\tau_e$ = 1.0 h, empty 510 μm ID tubing with a volume of 240 μl was inserted between the mixing section and the nozzle. After this extra volume was filled with mixed solution, pressure from the pumps was released (by diverting the flow from the pumps with a manual two-way valve[38]) so that the solution remained in the extra volume for the desired $\tau_e$ period. Pressure was then re-applied (by redirecting the flow to the mixer) to drive the mixed solution through the nozzle and freeze it on the cold copper surface.

To produce final solutions with 1.5 mM Aβ40 and 175 mM sodium phosphate at pH 7.4, 160 μl of 2.3 mM Aβ40 in 20 mM NaOH at pH 12 and 80 μl of 524 mM sodium phosphate buffer at pH 7.4 were pumped through the mixer with a 2:1 flow ratio. Samples with $\tau_e$ = 0 were prepared by rapidly mixing 160 μl of 2.3 mM Aβ40 in 20 mM NaOH, 20% v/v glycerol, and 10 mM sulfoacetyl-DOTOPA with 80 μl of 20 mM NaOH, 20% v/v glycerol, and 10 mM sulfoacetyl-DOTOPA at a total flow rate of 2.0 ml/min.

## DNP-enhanced ssNMR measurements

DNP-enhanced ssNMR measurements were performed at 9.4 T (100.8 MHz $^{13}$C NMR frequency) with a Bruker Avance III spectrometer console and Bruker Topspin 3.2 software. An extended interaction oscillator (Communications & Power Industries) and quasi-optical interferometer (Thomas Keating Ltd.). provided 1.5 W of a circularly polarized microwaves at 263.9 GHz to the home-built helium-cooled ssNMR probe, which has been described previously[41]. Measurements were performed at 7.00 kHz MAS frequency and 25 K sample temperatures, with a liquid helium consumption rate of 1.5–1.8 l/h, using MAS rotors with 4.0 mm outer diameters and 80 μl sample volumes. Assuming a 50% packing fraction, each ssNMR sample contained about 60 nmol of Aβ40 with [Aβ40] = 1.5 mM.

Before measurements on frozen Aβ40 solutions, the probe was cooled to 25 K while spinning a potassium bromide sample. The sample temperature was monitored by measuring the $^{79}$Br spin-lattice relaxation rate[69]. Rapidly frozen Aβ40 samples were transferred from liquid nitrogen to the probe without warming above approximately 100 K, by lowering the probe below the NMR magnet with the liquid helium transfer line remaining connected to the probe throughout the sample exchange operation.

For $^1$H-$^{13}$C cross-polarization, a $^1$H radio-frequency (RF) field of 54 kHz and a $^{13}$C RF field of 47 kHz was used. For $^1$H decoupling, a $^1$H RF field amplitude of 95 kHz was used with two-pulse phase modulation (TPPM)[70]. Double-quantum-filtered 1D $^{13}$C spectra were acquired with the RF pulse sequences utilizing the SPC5 recoupling technique[44]. In 2D $^{13}$C-$^{13}$C spectra, the $t_1$ increment was 40.0 μs and the maximum $t_1$ value was 8.0 ms. The recycle delay was determined to be 1.26 times the DNP build-up time ($\tau_{DNP}$). Values of $\tau_{DNP}$ were measured by fitting 1D saturation recovery data with single-exponential functions. Typically, $\tau_{DNP}$ was 3.0–4.0 s. DNP enhancement factors for cross-polarized $^{13}$C signals (microwaves on vs. microwaves off) were about 65 for rapidly frozen samples containing 20% v/v glycerol and approximately 100 for the more slowly frozen fibril sample containing 40% v/v glycerol. Spin diffusion mixing periods in 2D $^{13}$C-$^{13}$C spectra were 20 ms or 1.0 s to detect intra-residue or inter-residue crosspeaks, respectively.

2D $^{13}$C-$^{13}$C spectra were processed in nmrPipe[71] (version 9.4, Rev. 2017.335.16.23) with 100 Hz Gaussian apodization in both dimensions for spectra in Figs. 3a and 7a and Supplementary Figs. 3 and 9a, and 150 Hz Gaussian apodization in both dimensions for spectra in Figs. 4a and 7c and Supplementary Figs. 5, 7, and 9b. 2D spectra were plotted with nmrDraw (version 9.4, Rev. 2017.335.16.23) and Sparky

(version 3.114) software. Rmsd values in Figs. 3 and 4 and crosspeak volumes in Fig. 7 were calculated with Python scripts, after converting 2D spectral data to Python arrays using the NMRglue package[72] (version 0.8). In rmsd calculations, 2D spectra were first symmetrized about the diagonal and normalized to the total volume within the selected spectral regions. In rmsd calculations of aliphatic-aliphatic regions (Figs. 3b and 4b), signals within 5 ppm of the diagonal and within a square region between 50 ppm and 70 ppm were not included.

$^{13}$C chemical shifts are referenced to sodium trimethylsilylpropanesulfonate (DSS), using a value of 74.7 ppm for the natural-abundance glycerol $C_2$ signal in spectra of frozen Aβ40 solutions. This value was confirmed by direct measurements on aqueous DSS solutions containing 20% v/v and 40% v/v glycerol at room temperature. Measurements on a frozen solution at 100 K also showed a 74.7 ± 0.1 ppm difference between DSS and glycerol signals.

## Time-resolved light scattering

Time-resolved light scattering data were acquired with an Applied Photophysics SX20 stopped flow spectrometer and Applied Photophysics Pro-Data SX software (version 2.5.1852.0). Although this instrument is typically used for time-resolved fluorescence measurements, with different wavelengths for excitation and detection, time-resolved light scattering measurements are also possible when excitation and detection wavelengths are equal. For measurements in Fig. 5 and Supplementary Fig. 8, a Xe-Hg arc lamp and monochromator provided light at 562 nm wavelength for irradiating an optical cell with 20 μl volume. Scattered light was filtered through a single-band filter at 562 ± 20 nm (Semrock, part number FF01-561/14-25) and detected by a photomultiplier tube (PMT) mounted at 90-degree angle with respect to the incident light.

All solutions used for light scattering experiments were filtered with 0.2 μm syringe filters and degassed by stirring in a flask connected to the house vacuum. Solutions were additionally centrifuged at 80,000 × $g$ for 1.0 h to remove dust particles or aggregated material. Initially, the optical cell was flushed with the buffer solution until the PMT voltage reading reached its minimum value. Solutions to be mixed were loaded into 1.0 ml syringes, then rapidly mixed by the stopped flow instrument to make final solutions containing 1.5 mM (or 0.75 mM) Aβ40, 175 mM sodium phosphate, and 20% v/v glycerol at pH 7.4. Approximately 120 μl of total volume was injected through the optical cell in each scan. The PMT sampling time was set to 12.5 μs, and the numbers of samples for each data point were 20, 4800, and 32,000 for light scattering data with 0.5 s, 600 s, and 4000 s scan times, respectively. For measurements with a 0.5 s scan time, the measurements were repeated six times and averaged together.

## Stopped flow ThT fluorescence

Time-resolved ThT fluorescence data were acquired with the same stopped flow spectrometer used for the light scattering experiments. Fluorescence excitation light at 450 nm was provided by the same arc lamp and monochromator. Fluorescence emission was detected by the PMT after passing through a 488 nm long pass filter (Semrock, part number: BLP01-488R-25). After rapid mixing by the stopped flow instrument, solutions contained 25 μM ThT, 175 mM sodium phosphate, and 20% glycerol at pH 7.4, with Aβ40 concentrations from 29 μM to 1.5 mM.

## Circular dichroism spectroscopy

CD spectra in Supplementary Fig. 2 were recorded with a JASCO J-1500 spectrometer and JASCO Spectra Manager software (version 2.13.00), using a 0.2 mm path length cuvette containing 100 μM Aβ in either 20 mM NaOH at pH 12 or 175 mM sodium phosphate at pH 7.4. The spectra were acquired between 190 nm and 250 nm with a 1 nm interval, 2.0 s integration time, and 1 nm bandwidth.

**Reporting summary**

Further information on research design is available in the Nature Portfolio Reporting Summary linked to this article.

## Data availability

2D ssNMR spectra and TEM images generated in this study are available from Mendeley Data at https://doi.org/10.17632/kcjbbz9gzs.1. Light scattering data, circular dichroism data, and ThT data are provided in the Source data file. All other data are available from the authors upon request. Source data are provided with this paper.

## Code availability

Fortran95 code for simulating oligomer growth and light scattering signals and for fitting simulations to experimental time-resolved light scattering data are available from Mendeley Data at https://doi.org/10.17632/hkzth2dmms.1.

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

## Acknowledgements

This work was supported by the Intramural Research Program of the National Institute of Diabetes and Digestive and Kidney Diseases, National Institutes of Health. Support was provided to RT under project 1-ZIA-DK029029. We thank Dr. Kiyoshi Mizuuchi for assistance with time-resolved light scattering and fluorescence measurements.

## Author contributions

J.J. designed the research, prepared samples, performed measurements, analyzed data, and wrote the paper; W.-M.Y. synthesized and purified isotopically labeled peptide samples and triradical compounds; R.T. designed the research, analyzed data, performed simulations, and wrote the paper.

## Competing interests

The authors declare no competing interests.
