## [Peer Review File · Nature Communications]

REVIEWERS' COMMENTS

Reviewer #1 (Remarks to the Author):

The new data on the time-resolved structural properties of amyloid-beta(1-40) oligomerization utilizes a unique apparatus (described previously), which enables unprecedented time resolution of these difficult-to-characterize and functionally important species. The work is carried out in rigorous fashion and thorough analysis and is expected to have a very high impact in the field. I am strongly in favor of its acceptance to the journal. The description of solid-state NMR data and analysis is very clear and does not require any further refinement.

The following are suggestions for minor revisions of the manuscript.

- 1) In regards to the light-scattering data and their analysis, it would be helpful to qualify how the conclusions in the section "modeling the of oligomer growth as a coagulation process" may be affected if different shapes of the oligomers were taken into account. While the model is valid as is and is an adequate approximation, the differences in shapes seem like a significant factor that can have a sizeable quantitative impact on the parameters. For example, what happens if a distribution of shapes of different extent of ellipticities are introduced.
- 2) It also would be helpful to comment how these pathways and the structural features of the oligomers might be in principle connected to in vivo conditions. I.e., to what extent these results can be utilized to understand the pathological formation of oligomers.

Reviewer #2 (Remarks to the Author):

This manuscript submitted to Nature Communications by the Tycko lab entitled "Early events in amyloid- β self-assembly probed by time-resolved solid state NMR and light scattering" is exceptional and should be accepted with minor revisions.

The paper focuses on the intramolecular and intermolecular conformational changes in A β triggered by a pH jump from pH 12 to 7.4, employing a rapid mixing device capable of millisecond time resolution. This technological tour de force manuscript focusing on the early structures formed by A β when it is self-assembly competent is highly relevant because we still do not have a structure-proteotoxicity relationship in Alzheimer's disease (AD). In other words if A β is constantly populating assembled structures distinct from the parallel in register cross-beta-sheet amyloid fibrils then we have to consider whether these dynamic assemblies play a role in mediating neurodegeneration.

Specifically, the authors employ rapid mixing along with freeze quench coupled to solid state NMR to characterize the A β intramolecular and intermolecular conformational changes on the millisecond time scale. They then also do rapid mixing-based dynamic light scattering experiments to quantify time-dependent the quaternary structural changes (degree of A β self-assembly occurring as a function of time on the millisecond time scale. This technological tour de force combination of time resolved structural and solution MW data paints a new picture of how A β assembles. One- and two-dimensional ssNMR spectra presented by the authors indicate that β -strand conformations within individual monomeric A β polypeptides featuring contacts between the two main hydrophobic segments of A β 40 which develop within 1 ms after the pH drop, while light scattering data further support a primarily monomeric state up to 5 ms. Remarkably, as nonfibrillar oligomers grow to average sizes that exceed 100 molecules, conformational distributions and the overall level of molecular structural order remain nearly unchanged. Close intermolecular contacts between residues 18 and 33 develop within 500 milliseconds, when A β 40 is approximately octameric. Notably, and pertinent to my argument for rapid publication of this paper, these contacts argue against β -sheet organizations resembling those found previously in protofibrils and fibrils!

A striking feature of the Tycko data presented in Fig. 5b is the nearly linear increase in scattering signal beyond $t = 300$ s. In an attempt to explain this behavior, they considered a simple model for oligomer growth in which oligomers of size n and m can fuse irreversibly to form oligomers of size $n+m$, with rate constants $r_{n,m}$. Such a model describes a process that can be called "coagulation". Apparently, oligomer fusion occurs only rarely when oligomers collide with one another. This conclusion is consistent with TEM images, which show clusters of oligomers with various sizes, in contact with one another after adsorption and drying on the TEM grid but not fused, which supports why others see steady state $A\beta$ oligomer structures.

This paper is very well written, experimentally executed and argued. So why then do I request a very minor revision? Basically, Figure 1A is too small for me to see what is going on. I suggest converting figure 1A into a full page figure, so we can see what monomer structures are compatible with the SS-NMR data, ideally labeling a subset of residues in at least one monomer with residue numbers. Ditto with the oligomers, show us the details of what portions of $A\beta$ are self-assembling making the oligomer big enough to see, labeling the interacting sequences. Figure 1b, c and d could become Figures 2a, b, and c.

The Discussion nicely integrates this work with previous work.

REVIEWERS' COMMENTS

Reviewer #1 (Remarks to the Author):

The new data on the time-resolved structural properties of amyloid-beta(1-40) oligomerization utilizes a unique apparatus (described previously), which enables unprecedented time resolution of these difficult-to-characterize and functionally important species. The work is carried out in rigorous fashion and thorough analysis and is expected to have a very high impact in the field. I am strongly in favor of its acceptance to the journal. The description of solid-state NMR data and analysis is very clear and does not require any further refinement.

We thank the reviewer for these favorable comments.

The following are suggestions for minor revisions of the manuscript.

1) In regards to the light-scattering data and their analysis, it would be helpful to qualify how the conclusions in the section "modeling the of oligomer growth as a coagulation process" may be affected if different shapes of the oligomers were taken into account. While the model is valid as is and is an adequate approximation, the differences in shapes seem like a significant factor that can have a sizeable quantitative impact on the parameters. For example, what happens if a distribution of shapes of different extent of ellipticities are introduced.

Since the diameters of the oligomers in our experiments are much less than the wavelength of light in our light scattering measurements, the oligomer shape has only a weak effect on the scattered light intensity. This is because any dependence on shape arises from destructive interference between light rays that are scattered from different points within the oligomer. Destructive interference is a small effect because the maximum phase difference between such light rays can only be as large as $360^\circ \times (\text{diameter/wavelength}) \times (\text{refractive index}) \approx 10^\circ$.

To quantify this assertion, we have added the following sentence at the end of the section entitled *Evolution of oligomer sizes from time-resolved light scattering*:

"To be specific, for randomly oriented spheroidal particles with 524 nm^3 volume (10 nm diameter if spherical), the scattering intensity perpendicular to the incident light beam is calculated⁴⁷ to vary by only 3% as the aspect ratio of the particles varies between 0.3 (oblate) and 3.0 (prolate)."

The calculations are based on the Rayleigh-Debye light scattering theory described in ref. 47, which is a classic textbook on the subject.

2) It also would be helpful to comment how these pathways and the structural features of the oligomers might be in principle connected to in vivo conditions. I.e., to what extent these results can be utilized to understand the pathological formation of oligomers.

To address this point, we have added the following paragraph to the Discussion section:

“As with previous *in vitro* studies of aggregation mechanisms, not all results from our time-resolved ssNMR and light scattering experiments are expected to be directly relevant to A β self-assembly *in vivo*, since the peptide concentrations in our experiments are relatively high, the maximum time is relatively short, and other biochemical components and biological processes are not included. Aspects that may be relevant to self-assembly *in vivo* include the observed preference for β -strand conformations at the monomer level, the observed difference in oligomer fusion rates for oligomer sizes below and above $n \approx 16$, and the absence of ordered β -sheet structures within the oligomers, even when $n_{ave} > 100$.”

Reviewer #2 (Remarks to the Author):

This manuscript submitted to Nature Communications by the Tycko lab entitled “Early events in amyloid- β self-assembly probed by time-resolved solid state NMR and light scattering” is exceptional and should be accepted with minor revisions.

The paper focuses on the intramolecular and intermolecular conformational changes in A β triggered by a pH jump from pH 12 to 7.4, employing a rapid mixing device capable of millisecond time resolution. This technological tour de force manuscript focusing on the early structures formed by A β when it is self-assembly competent is highly relevant because we still do not have a structure-proteotoxicity relationship in Alzheimer’s disease (AD). In other words if A β is constantly populating assembled structures distinct from the parallel in register cross-beta-sheet amyloid fibrils then we have to consider whether these dynamic assemblies play a role in mediating neurodegeneration.

Specifically, the authors employ rapid mixing along with freeze quench coupled to solid state NMR to characterize the A β intramolecular and intermolecular conformational changes on the millisecond time scale. They then also do rapid mixing-based dynamic light scattering experiments to quantify time-dependent the quaternary structural changes (degree of A β self-assembly occurring as a function of time on the millisecond time scale. This technological tour de force combination of time resolved structural and solution MW data paints a new picture of how A β assembles. One- and two-dimensional ssNMR spectra presented by the authors indicate that β -strand conformations within individual monomeric A β polypeptides featuring contacts between the two main hydrophobic segments of A β 40 which develop within 1 ms after the pH drop, while light scattering data further support a primarily monomeric state up to 5 ms. Remarkably, as nonfibrillar oligomers grow to average sizes that exceed 100 molecules, conformational distributions and the overall level of molecular structural order remain nearly unchanged. Close intermolecular contacts between residues 18 and 33 develop within 500 milliseconds, when A β 40 is approximately octameric. Notably, and pertinent to my argument for rapid publication of this paper, these contacts argue against β -sheet organizations resembling those found previously in protofibrils and fibrils!

A striking feature of the Tycko data presented in Fig. 5b is the nearly linear increase in scattering signal beyond $t = 300$ s. In an attempt to explain this behavior, they considered a simple model for oligomer growth in which oligomers of size n and m can fuse irreversibly to form oligomers of size $n+m$, with rate constants $r_{n,m}$. Such a model describes a process that can be called

"coagulation". Apparently, oligomer fusion occurs only rarely when oligomers collide with one another. This conclusion is consistent with TEM images, which show clusters of oligomers with various sizes, in contact with one another after adsorption and drying on the TEM grid but not fused, which supports why others see steady state A β oligomer structures.

This paper is very well written, experimentally executed and argued. So why then do I request a very minor revision? Basically, Figure 1A is too small for me to see what is going on. I suggest converting figure 1A into a full page figure, so we can see what monomer structures are compatible with the SS-NMR data, ideally labeling a subset of residues in at least one monomer with residue numbers. Ditto with the oligomers, show us the details of what portions of Abeta are self-assembling making the oligomer big enough to see, labeling the interacting sequences. Figure 1b, c and d could become Figures 2a, b, and c.

The Discussion nicely integrates this work with previous work.

We thank the reviewer for these favorable comments. To address the reviewer's request for a revised Figure 1, we have enlarged panel a of Figure 1 relative to panels b, c, and d. However, we have not labeled residues or interacting sequences. This is because Figure 1a is only a conceptual illustration of the process of oligomer formation, intended to accompany the Introduction section and help readers understand the questions that motivate the work described in this manuscript. The figure legend says: "Hypothetical depiction of A β 40 self-assembly". Monomer and oligomer structures in Figure 1a are not constrained by ssNMR data.